Methods

# Efficient identification of de novo mutations in family trios: a consensus-based informatic approach

Mariya Shadrina[1], Özem Kalay[2], Sinem Demirkaya-Budak[2], Charles A LeDuc[3], Wendy K Chung[4], Deniz Turgut[2], Gungor Budak[2], Elif Arslan[2], Vladimir Semenyuk[2], Brandi Davis-Dusenbery[2], Christine E Seidman[5,6], H Joseph Yost[7], Amit Jain[2], Bruce D Gelb[1,8]

Accurate identification of de novo variants (DNVs) remains challenging despite advances in sequencing technologies, often requiring ad hoc filters and manual inspection. Here, we explored a purely informatic, consensus-based approach for identifying DNVs in proband–parent trios using short-read genome sequencing data. We evaluated variant calls generated by three sequence analysis pipelines—GATK HaplotypeCaller, DeepTrio, and Velsera GRAF—and examined the assumption that a requirement of consensus can serve as an effective filter for high-quality DNVs. Comparison with a highly accurate DNV set, validated previously by manual inspection and Sanger sequencing, demonstrated that consensus filtering, followed by a force-calling procedure, effectively removed false-positive calls, achieving 98.0–99.4% precision. At the same time, sensitivity of the workflow based on the previously established DNVs reached 99.4%. Validation in the HG002-3-4 Genome-in-a-Bottle trio confirmed its robustness, with precision reaching 99.2% and sensitivity up to 96.6%. We believe that this consensus approach can be widely implemented as an automated bioinformatics workflow suitable for large-scale analyses without the need for manual intervention, especially when very high precision is valued over sensitivity.

## Introduction

Germline de novo variants (DNVs) play a crucial role in evolution, introducing new genetic variation. At the same time, DNVs underlie a wide range of genetic diseases, increasing the interest in studying the frequency and characteristics of sporadic mutations in human genomes (Acuna-Hidalgo et al, 2016; Deciphering Developmental Disorders Study, 2017; Goldmann et al, 2019). With the recent availability of genome sequencing (GS), genetic studies of trios consisting of an affected proband and unaffected parents provide a direct method for the large-scale detection of DNVs (Nicolas & Veltman, 2019; Richter et al, 2020). Although the genome sequence of an individual can differ at 4–5 million positions compared with the human reference genome (Auton et al., 2015), the vast majority of the observed genetic variation is inherited. The germline de novo mutation rate for single nucleotide variants (SNVs) in human genomes is estimated as $1.0-1.8 \times 10^{-8}$ per nucleotide per generation, which manifests as 44–82 de novo SNVs for an individual (including one to two variants in coding regions) and is dependent upon parental ages, predominantly paternal age (Acuna-Hidalgo et al, 2016; Goldmann et al, 2019). In addition to SNVs, only three to nine small de novo insertions/deletions (indels), which are typically shorter than 50 bp, are expected per human genome. As a result, the prior odds of a variant observed only in the proband genome being a DNV remain modest. Outnumbered by inherited variants, detection of DNVs is a nontrivial task, resulting in many false-positive variant calls, especially in regions of low coverage or with high levels of noise.

In our previous work, we studied 763 probands with congenital heart disease (CHD) and their unaffected parents with trio GS (Richter et al, 2020). We identified 71 de novo SNVs and five de novo indels per CHD proband on average, corresponding to expected rates of true de novo SNVs and indels around 98% and 94%, respectively (based on PCR-based Sanger sequencing). However, accurate detection of DNVs with high precision and sensitivity was achieved using a sophisticated workflow that included manual inspection of ambiguous variants. This limits the scalability of that method for studies of larger cohorts with trio GS, which are becoming increasingly commonplace as costs have decreased. Here, we report the development of a fully automated trio GS workflow implementing three independent pipelines, Broad Institute's Best Practices Pipeline for Germline Short Variant Discovery (GATK4) (DePristo et al, 2011), Velsera GRAF Germline Variant Detection

---

[1]Mindich Child Health and Development Institute and the Department of Genetics and Genomic Sciences, Icahn School of Medicine, New York, NY, USA   [2]Velsera Inc., Charlestown, MA, USA   [3]Department of Pediatrics, Columbia University, New York, NY, USA   [4]Department of Pediatrics, Boston Children's Hospital, Harvard Medical School, Boston, MA, USA   [5]Division of Cardiovascular Medicine, Brigham and Women's Hospital, Harvard Medical School, Boston, MA, USA   [6]Howard Hughes Medical Institute, Chevy Chase, MD, USA   [7]Molecular Medicine Program, University of Utah, Salt Lake City, UT, USA   [8]Department of Pediatrics, Icahn School of Medicine, New York, NY, USA

Correspondence: bruce.gelb@mssm.edu

Workflow (GRAF) (Rakocevic et al, 2019), and BWA-DeepTrio (Kolesnikov et al, 2021 *Preprint*), to accurately call DNVs. We included the pangenome-aware GRAF workflow in our consensus panel because it was demonstrated to increase the recall of short variants in Genome-in-a-Bottle (GIAB) benchmark samples sequenced on the Illumina short-read platform (Rakocevic et al, 2019; Olson et al, 2022) and thus enabled further analysis of our previously processed dataset with the possibility of discovering DNVs missed by the GATK4 method.

# Results

### The first QC step

GS data from 10 parent–offspring trios from the Pediatric Cardiac Genetics Consortium (PCGC) database were analyzed. Each trio consisted of an individual with CHD and their healthy parents. To analyze the GS trio data, we ran three analytic pipelines to call de novo SNVs and indels. All steps performed in the analysis are shown in Figs 1, S1, S2A and B, and S3 and Table S1.

At the first QC step, we applied hard-threshold filtering using the variant annotations from joint VCFs, to eliminate low-quality calls from GATK4 and DeepTrio outputs. For GRAF outputs, we followed a similar hard-thresholding step using annotations from merged VCF and read alignments, after handling representation differences between variants from each family member (Supplemental Data 1). The details of annotations and thresholds used for each pipeline are in Tables S2 and S3.

The resulting candidate DNVs after the initial filtering steps comprised 1,409 SNVs and 3,915 indels for GATK4, 4,717 SNVs and 14,914 indels for DeepTrio, and 21,795 SNVs and 2,546 indels for GRAF. The number of filtered Mendelian-inconsistent variant calls at this stage exceeds the expected count of DNVs in 10 probands by at least an order of magnitude (Acuna-Hidalgo et al, 2016). In addition, these variants showed enrichment in indels compared with the background distribution because of the less accurate calling of indels, which resulted in a higher number of false-positive indel calls, as well as because of our filtering method's aggressive removal of irrelevant SNPs.

### Regional and population filters

After the initial filtering steps, candidate variants from the three pipelines were further refined with regional and population filters. The regional filter removed variants located in low-complexity regions, low-mappability regions (Karimzadeh et al, 2018), EN-CODE blacklists (ENCODE Project Consortium, 2012), and segmental-duplication regions (Vollger et al, 2022). The population filter removed all variants with allele frequencies > 0.1% based on the gnomAD exome (v2.1.1) (Karczewski et al, 2020), gnomAD genome (v2.1.1) (Karczewski et al, 2020), and 1,000 Genome (Katsnelson, 2010) databases, as variants with high frequency are unlikely to be pathogenic for most Mendelian traits. The final GATK4 candidate DNVs included 704 SNVs and 643 indels; DeepTrio candidate DNVs included 803 SNVs and 1,260 indels; and the GRAF candidate DNVs included 2 812

SNVs and 272 indels (Fig 1, central). The union set of DNVs from all three workflows contained 3,120 SNVs and 2,071 indels.

### Consensus step

After regional and population filters, we observed that almost all high-confidence DNVs from the previous work (Richter et al, 2020) (Freeze variants) were called by at least two pipelines. Given that our primary focus in this work was to enhance precision in de novo calling, we discarded all variants identified by a single method. As a result, 634 of 3,120 SNVs and 62 of 2,071 indels were retained (Fig 1, central, Fig 2A and B), a total of 696 putative DNVs across the 10 PCGC trios (i.e., 69.6 DNVs/proband).

### Ambiguous variants after the consensus step

To evaluate performance of the consensus approach, we performed a visual inspection of the 696 candidate DNVs using BAM files, which revealed that 61 variants had a more complicated read composition than regular de novo SNVs and indels and could be inherited or result from alignment errors (Fig 1, right). As the primary goal of our study was to reach a high precision in calling DNVs, we explored the characteristics of these variants to develop additional filtering criteria.

### *Alternative alleles in homozygous variants in parents (AAHP filter)*
Of the 61 ambiguous variants found after the consensus step, 41 variants had alternate allele-carrying reads in parents' pileups even though the variant calls were assigned as homozygous reference in the parents by a variant caller (Fig 1, right, Fig S4A). We considered that many of these variants could have resulted from alignment errors, where alternative alleles having a lower alignment score than the corresponding reference alleles were partially missed by an aligner in a parent. However, a single read in a parent alignment showing an alternate SNV coinciding with a de novo mutation in the proband may result from technical errors commonly associated with the sequencing process. Considering the latter cases, we applied a threshold for alternative allele-carrying reads (AAC) ≤ 1 for SNVs and 0 for indels in the parent samples. Although these reads could also indicate low-level parental mosaicism, we retained them because that would still be consistent with high-impact variants of clinical significance (Cook et al, 2021). Based on the AAHP thresholds, we expect 10 of the 41 variants to be removed as false positives. The PCR validation we describe below in the section "Sanger Sequencing Confirmations" confirmed the rationale of the AAHP filter.

### *Proband haplotype variants (PH filter)*
We found that 25 of the 61 ambiguous variants (including five variants that had alternate alleles in parents as well) belong to de novo clusters, a notable concentration of de novo SNVs within a relatively confined genomic region (Fig 1, right, Fig S4B). The median length of the observed DNV concentrations is seven base pairs, whereas the smallest and largest clusters correspond to 2 and 57 base pairs, respectively. These clusters consist of 2–8 SNVs, with an average inter-SNV distance of three base pairs. We considered that such occurrences might stem from alignment errors, as a substantial number of multiple proximate de novo events are unlikely. On the other hand, some neighboring de novo mutations might be

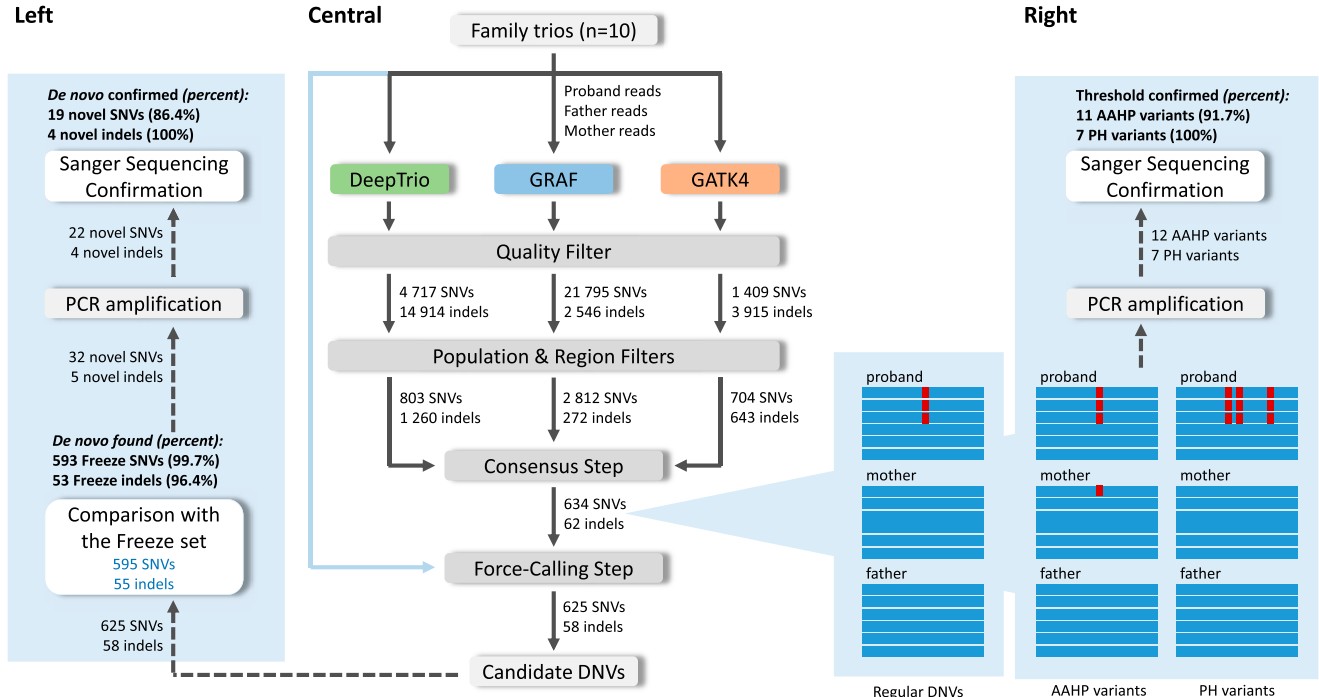

**Figure 1. Three pipelines were applied to analyze 10 PCGC trios: GATK4, GRAF, and DeepTrio.**
Three independent sets of possible de novo variants in probands were found for each family, then filtered with regional and population filters. Only variants found by at least two pipelines were retained. At the final step, the force-calling filter was performed (central). The final list of candidate DNVs was compared with the Freeze set, and novel DNVs underwent Sanger sequencing confirmation (left). Thresholds for the force-calling filter were also confirmed with the Sanger sequencing (right). The threshold for variants with alternative alleles in homozygous variants in parents was assigned as AAC ≤ 1 for SNVs and 0 for indels in the parent samples, and the threshold for proband haplotype variants was confirmed as 20 bp. Examples of the alternative alleles in homozygous variants in parents and proband haplotype variants are shown in Fig S5.

synchronous within a region of up to 20 base pairs, as discussed previously (Brand et al, 2024). PCR validation conducted in this study, detailed in the section "Sanger Sequencing Confirmations," confirmed the presence of DNV clusters ranging from 2 to 20 base pairs in length. Therefore, we implemented a filtering process to exclude any concentration of DNVs that extend beyond 20 base pairs. Based on the PH threshold, we expect four of the 25 variants to be removed as false positives.

## Force-calling step

Although we observed the AAHP and PH variants in BAM files—using Integrative Genomics Viewer (IGV) (Thorvaldsdóttir et al, 2013)—we needed a better way to find such calls—an automated filter, which can be applied for large-scale analyses without the need for manual intervention. We found that using force-calling with HaplotypeCaller from the GATK4 pipeline effectively fulfilled this purpose. By default, HaplotypeCaller performs variant calling in regions that have evidence for genomic variation, termed active regions. After determining that the region is active, HaplotypeCaller reassembles the reads, creates a sequence for the region, and determines the allele. HaplotypeCaller also allows for reassembling the reads at a given region even if the region is not active (–force-call runtime parameter). We used this functionality to conduct a more thorough examination of candidate de novo variants

identified during the consensus step and to search for variant evidence in parents. In this work, we will refer to this step as "force-calling." For the PH filter, we used the haplotype information provided by HaplotypeCaller as part of its variant calling process.

To remove the 14 ambiguous variants (Table 1), we recalled the 696 candidate DNVs in corresponding families with HaplotypeCaller from the GATK4 pipeline. After the application of the force-calling step, we implemented a second round of quality filtering and applied AAHP and PH filters. Table S2 shows additional QC filters applied to the candidate variants after the force-calling step. After this step, the total number of DNVs decreased to 683, whereas the 13 ambiguous variants were automatically removed (Table 1). The last ambiguous variant was missed by our PH filter as HaplotypeCaller could not correctly determine all related variants and the haplotype length was estimated as 10 instead of 57 base pairs (Fig S6A). Importantly, the force-calling step takes less than 15 CPU minutes per trio and is easily scalable for large datasets.

## Comparison of the three pipelines

Most of the final candidate DNV sets were found by all three methods: 563 SNVs and 50 indels (Fig 2C and D). However, the GRAF pipeline added 18 and 47 DNVs when overlapped with GATK4 and DeepTrio, respectively. In contrast, the consensus between GATK4 and DeepTrio added only five DNVs.

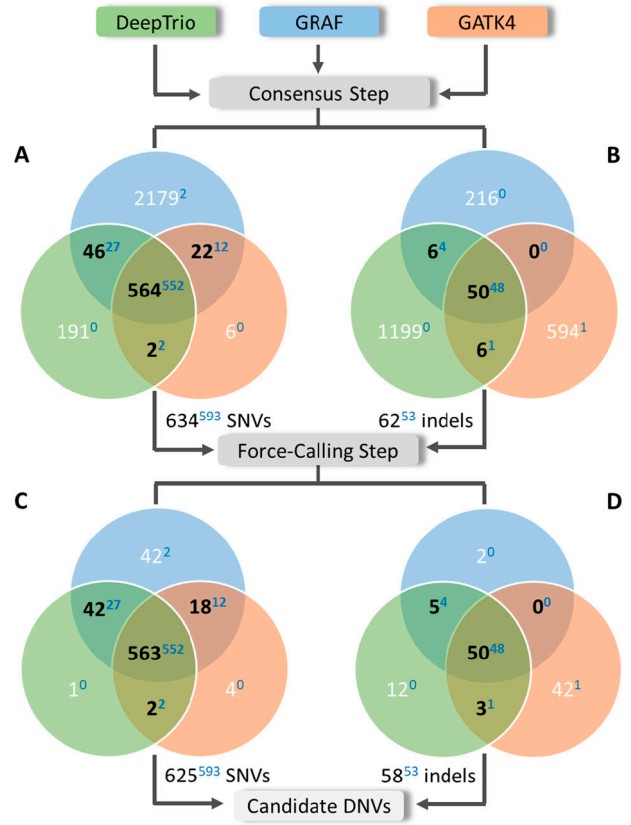

**Figure 2. The de novo candidates in the 10 PCGC trios after the consensus and force-calling steps.**
**(A, B, C, D)** Distribution of the de novo candidates in the 10 PCGC trios after the consensus step (A, B) and after the force-calling step (C, D) between the three pipelines: (A, C) SNVs and (B, D) indels. Only variants found by at least two methods (black) were included in further analyses, resulting in 625 SNVs and 58 indels as a final list of candidate DNVs. Numbers of variants found by a single method are colored in white, whereas numbers of Freeze variants are shown as blue superscript.

The consensus workflow combining the results from the three orthogonal variant calling methods showed the best results with the largest total number of DNVs (683 variants), though it was also the most computationally expensive to run (Table 1). The pipeline with all three methods on average required 4,644 CPU hours to run on a computer with 72 Intel Xenon CPUs. Using GRAF and DeepTrio together identified 660 DNVs(missing 3.4% of variants from the three-method set) but only required 2,484 CPU hours per trio, whereas the GATK4 and DeepTrio combination was the least effective, identifying only 618 DNVs (missing 9.5% of DNVs found by the three-method option) while requiring 3,348 CPU hours. Combining GATK4 and GRAF revealed 631 DNVs (missing 7.6% of the three-method set) and taking 3,456 CPU hours per trio. Figs S1 and S5 show the CPU hour distribution over different pipelines.

## Comparison with the Freeze set

The previous PCGC study called DNVs with a combination of GATK, FreeBayes, and a convolutional neural network trained on manually curated IGV plots (Richter et al, 2020). That pipeline found 752

variants for the 10 trios (Freeze set), with a true DNV call rate of > 95% based on Sanger sequencing confirmation of a modest number of DNV calls. We reassessed those Freeze set variants by applying regional and population filters, which narrowed the DNV set to 658 variants. Eight variants did not pass visual verification of the BAM files in IGV and were removed as FPs. The remaining 650 Freeze variants were considered as true DNVs and used for comparison (Fig 1, left).

Combining all three pipelines (GATK4, DeepTrio, and GRAF) followed by the force-calling step detected 646 of the 650 Freeze variants (Fig 2C and D, Table 1). Two Freeze set de novo SNVs were called only by the GRAF pipeline and were excluded in the consensus step, and one Freeze set de novo indel was called only by the GATK4 method. Another Freeze set de novo indel was missed by all three pipelines. In summary, combining all three pipelines and using the force-calling step identified 646 Freeze DNVs (99.4%) but called additional 37 DNVs (5.7% increment) (Table 1).

## Sanger Sequencing Confirmations

A subset of candidate DNVs underwent validation in the proband and both parents with Sanger sequencing of amplicons after PCR amplification using primers designed within 100–400 bp of the variant (Zaidi et al, 2013). We added the 37 novel DNVs identified in this study to the subset (Fig 1, left). Also, we included both FP and TP variants based on the assigned thresholds for the AAHP and PH filters (Fig 1, right). In total, 92 variants were chosen for the Sanger sequencing confirmation of the novel DNVs, and to investigate the empirical thresholds, we developed for the force-calling step. Because many of the regions harboring these variants were complex, designing successful PCR assays proved technically challenging. As a result, we were only able to amplify sequences for 73 of our 92 target variants.

### Novel DNVs

Our three-method workflow revealed 37 novel de novo variants, which were not identified by the previous PCGC pipeline (Richter et al, 2020). 36 of the 37 variants were posited as de novo based on visual inspection of the BAM files in IGV. The other variant was assigned as PH with a concentration of de novo SNVs in a region of 57 base pairs that HaplotypeCaller failed to recognize as a haplotype (Fig S6A). Of the 37 novel de novo candidates, 26 variants were successfully amplified with PCR. Of those 26, 23 variants were confirmed as de novo. Two variants were not found in the proband, and one variant appeared to be inherited (Tables 2 and S4, Fig S6B).

### AAHP

We assigned upper thresholds for AAC of parents as 1 for SNVs and 0 for indels to remove variants caused by sequencing errors at the force-calling step. 12 variants, both SNVs and indels, that had a small number of alternate alleles (1–3) in parents were successfully amplified with PCR (Table S5). The consistency of the applied thresholds was confirmed for 11 variants (Table 3). Four variants were proved as inherited from a parent, whereas two variants, initially labeled as inherited, were verified as reference alleles in the proband. One variant, initially marked as inherited from the mother, was confirmed to have been inherited from the father,

**Table 1. Performance and computational costs for different combinations of the three pipelines.**

| Step | Methods | Candidate DNVs | | Freeze DNVs[a] | | Novel DNVs | | | Computational costs | |
|------|---------|-------|------------|-------|--------|-------|-------------------|-------------------|--------------|------------------------------|
| | | Calls | Ambiguous[b] | Found | Missed | Total | PCR-confirmed TP | PCR-confirmed FP | CPU hours | Runtime per trio[c] (hours) |
| Consensus | GATK4/GRAF/DeepTrio | 696 | 14 | 646 | 4 | 50 | 24 | 4 | 4,644 | 25 |
| Force-calling | GATK4/GRAF/DeepTrio | 683 | 1 | 646 | 4 | 37 | 23 | 3 | 4,644 | 25 |
| | GATK4/GRAF | 631 | 1 | 612 | 38 | 19 | 13 | 3 | 3,456 | 17 |
| | GATK4/DeepTrio | 618 | 0 | 603 | 47 | 15 | 12 | 2 | 3,348 | 25 |
| | GRAF/DeepTrio | 660 | 0 | 631 | 19 | 29 | 22 | 2 | 2,484 | 23.5 |

[a]The Freeze variants found previously (Richter et al, 2020) are updated according to the current regional and population filters resulting in 650 DNVs in total.
[b]Candidate DNVs assigned as AAHP or PH based on visual verification of the BAM files in Integrative Genomics Viewer (Thorvaldsdóttir et al, 2013) that fail to meet the established thresholds (see "Ambiguous Variants After Consensus Step").
[c]Estimated time required for running pipelines for a trio on the AWS cloud.

despite the absence of allele-supporting reads at the location (Table S5).

Only one variant, where maternal alignment had the three reads with an alternate allele, was found as de novo. Therefore, the PCR results showed that our parental AAHP filter was efficient for removing FP variants (Fig 1, right, Table 3).

### Proband haplotype variants

We considered cluster DNVs in probands as FP if the furthest mutations within a haplotype were located ≥ 20 bp apart. Seven variants with a haplotype in probands were successfully amplified with PCR (Tables 3 and S6). Four variants confirmed as TPs had a cluster length of 2, 4, 7, and 11 bp. Another three variants with a cluster length of 20, 29, and 31 bp were confirmed as FPs. Therefore, PCR validated the PH filter for all seven variants tested (Fig 1, right, Table 3). These results agree with the previously published PH threshold distance of 20 bp.

### Overview of de novo variants

We calculated the relative frequencies of mutation classes (Table 4A and B), which are in good accordance with mutation spectra previously published (Sasani et al, 2019). Of the 683 DNVs we found, there were 60 variants from exonic, UTR, and ncRNA regions, 266 intronic DNVs, and 348 intergenic ones (Table 4C).

### GIAB trio results

We repeated our workflow using the same parameters and thresholds for the HG002-3-4 Ashkenazi trio, gold-standard GIAB family with the established truth set of DNVs for high-confidence regions. After applying population and regional filters, the truth set resulted in 936 de novo SNVs and 55 de novo indels, which we further used for comparison (Fig S7, left).

The three methods (DeepTrio, GRAF, and GATK4) were run for the trio, and after the consensus step, 900 SNVs and 49 indels were found by at least two methods (Fig S7, central, S8AB). As for the 10 PCGC trios, most of DNVs for the HG002-3-4 trio were called by all three methods (556 SNVs and 23 indels). However, GATK4 found a

much smaller number of DNVs (601 variants) compared with DeepTrio (959 variants) and GRAF (1,246 variants) because of a high number of false-negative variants during the Genotype Posterior step (Table 5). Despite the low performance of GATK4, consensus-only results of the three methods revealed 892 of 936 truth SNVs (sensitivity = 95.3%) and 49 of 55 truth indels (sensitivity = 85.5%) (Fig S8A and B). The eight additional SNVs, which were not part of the truth set, were considered as false positives. This allowed us to estimate precision for the HG002-3-4 trio's results as 99.1% and 100% for SNVs and indels, respectively.

The force-calling step applied to the consensus list of DNVs filtered out only two indels, resulting in 900 SNVs and 47 indels in total (Fig S8C and D). As we called DNVs for the HG002-3-4 trio in high-confidence regions, we observed far fewer false-positive variants than we found in the PCGC trios (Fig 2 versus Fig S8), and therefore, the force-calling step did not improve the results. Notably, the force-calling step did remove many false-positive calls found by single methods only. For example, 273 false-positive and 973 true-positive variants were found by GRAF (Table 5, Fig S8A). After the force-calling step, 16 false-positive (5.7%) and 958 true-positive (98.5%) variants remained (Table 5, Fig S7, right), demonstrating the efficacy of this step for both high- and low-confidence calls.

We validated the performance of the AAHP and PH filters for the GIAB trio similarly as we did for the PCGC trios. After the consensus step, we selected candidate DNVs with alternate alleles in parents, 54 in total, and compared them with the truth set (Fig S7, right). The threshold for the AAHP filter assigned as AAC ≤ 1 for SNVs and 0 for indels in the parent samples was confirmed for 48 of the 54 variants (88.9%). 21 candidate DNVs after the consensus step had a haplotype in probands (Fig S7, right). We used the variants to evaluate the PH filter with the threshold assigned as 20 bp. Comparison with the truth set confirmed the accuracy of the threshold for 19 of the 21 variants (90.5%).

We estimated the total precision and sensitivity for the HG002-3-4 trio after running the full workflow as 99.2% and 94.8%, respectively (Table 5). The eight novel SNVs (0.8%) found in this work are presumably false-positive. Three of the eight variants had alternative alleles in parents and could be filtered out by a more rigorous filtering algorithm (Fig S6C). A visual inspection of the other five variants with IGV did not reveal any signs of them being misaligned or

**Table 2. Novel DNV confirmations.**

| | Assigned as DNV | PCR-amplified | Confirmed as | | |
|---|---|---|---|---|---|
| | | | DNV | Reference allele | Inherited |
| Novel DNVs | 37 | 26 | 23 | 2 | 1 |

**Table 3. Filter performance.**

| | Assigned as | PCR-amplified | Confirmed as | | |
|---|---|---|---|---|---|
| | | | DNV | Reference allele | Inherited |
| AAHP filter (SNPs: AAC ≤ 1 indels: AAC = 0) | DNV | 5 | 5 | 0 | 0 |
| | inherited/misaligned | 7 | 1 | 2 | 4 |
| PH filter (cluster length ≤ 20 bp) | DNV | 4 | 4 | 0 | 0 |
| | inherited/misaligned | 3 | 0 | 3 | 0 |

**Table 4. Statistics of de novo variants found in the three-method workflow.**

| A. Mutation class | Count | Average per sample | Relative frequency |
|---|---|---|---|
| SNP | 625 | 62.5 | 0.92 |
| SNP coding | 4 | 0.4 | 0.01 |
| Indel | 58 | 5.8 | 0.08 |
| Indel coding | 0 | 0 | 0 |
| **B. Mutation class** | **Count** | **Average per sample** | **Relative frequency** |
| C>T | 244 | 24.4 | 0.39 |
| CpG>TpG | 76 | 7.6 | 0.12 |
| C>A | 69 | 6.9 | 0.11 |
| CpG>ApG | 7 | 0.7 | 0.01 |
| C>G | 65 | 6.5 | 0.1 |
| CpG>GpG | 7 | 0.7 | 0.01 |
| T>C | 159 | 15.9 | 0.25 |
| T>G | 47 | 4.7 | 0.08 |
| T>A | 41 | 4.1 | 0.07 |
| **C. Mutation region** | **Count** | **Average per sample** | **Relative frequency** |
| Exonic | 4 | 0.4 | 0.01 |
| ncRNA | 47 | 4.7 | 0.07 |
| UTR3/UTR5 | 9 | 0.9 | 0.01 |
| Intronic | 266 | 26.6 | 0.39 |
| Downstream/upstream | 9 | 0.9 | 0.01 |
| Intergenic | 348 | 34.8 | 0.51 |

inherited. Discussion of the novelty/veracity of the eight DNVs found is, however, outside the scope of the present study.

# Discussion

The exploration of DNVs in a proband with any given trait by comparing that person's genome sequence with those of the unaffected parents is conceptually straightforward. However, identification of less than 100 DNVs among millions of inherited variants is challenging (Acuna-Hidalgo et al, 2016). Because of sequencing and alignment errors, some variants can be wrongly identified as de novo.

Significant efforts have been devoted to improving the efficiency of DNV detection. Although some methods have explored the use of variant annotations, specifically genotype likelihoods, from probands' and parents' samples to calculate confidence values for

**Table 5. Performance of different methods for the HG002-3-4 trio.**

| | Calls | TP | FP | FN | Sensitivity, % | Precision, % |
|---|---|---|---|---|---|---|
| **Total TP = 1,323 (Khazeeva et al, 2022)** | | | | | | |
| DeNovoCNN (threshold = 0.5) | 1,233 | 1,198 | 35 | 125 | 90.6 | 97.2 |
| DeNovoGear (threshold = 0.5) | 1,346 | 1,063 | 283 | 260 | 80.4 | 79.0 |
| DeNovoGear (threshold = 0.9) | 1,161 | 1,047 | 114 | 276 | 79.1 | 90.2 |
| DeepTrio (DeepVariant unfiltered, BQSR) | 1,207 | 1,127 | 80 | 196 | 85.2 | 93.4 |
| GATK4 | 1,338 | 1,171 | 167 | 152 | 88.5 | 87.5 |
| **Total TP = 995 (Brand et al, 2024)** | | | | | | |
| DeNovoCNN (threshold = 0.5) | 1,147 | 965 | 182 | 30 | 97.0 | 84.1 |
| modified DeepTrio (threshold = 0.985) | 1,041 | 973 | 68 | 22 | 97.8 | 93.5 |
| **Total TP = 991 (current study)** | | | | | | |
| DeepTrio | 959 | 935 | 24 | 56 | 94.3 | 97.5 |
| DeepTrio (force-calling) | 944 | 935 | 9 | 56 | 94.3 | 99.0 |
| GRAF | 1,246 | 973 | 273 | 18 | 98.2 | 78.1 |
| GRAF (force-calling) | 974 | 958 | 16 | 33 | 96.7 | 98.4 |
| GATK4 | 601 | 590 | 11 | 401 | 59.5 | 98.2 |
| GATK4 (force-calling) | 595 | 588 | 7 | 403 | 59.3 | 98.8 |
| GATK4 (no posteriors) | 1,027 | 951 | 76 | 40 | 96.0 | 92.6 |
| GATK4 (force-calling, no posteriors)[a] | 968 | 941 | 27 | 50 | 95.0 | 97.2 |
| DeepTrio/GRAF (force-calling) | 935 | 927 | 8 | 64 | 93.5 | 99.1 |
| DeepTrio/GATK4 (force-calling) | 583 | 580 | 3 | 411 | 58.5 | 99.5 |
| GATK4/GRAF (force-calling) | 587 | 584 | 3 | 407 | 58.9 | 99.5 |
| DeepTrio/GRAF/GATK4 (consensus only) | 949 | 941 | 8 | 50 | 95.0 | 99.2 |
| DeepTrio/GRAF/GATK4 (force-calling)[b] | 947 | 939 | 8 | 52 | 94.8 | 99.2 |
| DeepTrio/GRAF/GATK4 (consensus only, no posteriors)[a] | 966 | 957 | 9 | 34 | 96.6 | 99.1 |
| DeepTrio/GRAF/GATK4 (force-calling, no posteriors)[a] | 961 | 952 | 9 | 39 | 96.1 | 99.1 |

[a]The GATK4 pipeline was performed without the CalculateGenotypePosteriors step (Table S1).
[b]The full workflow we applied to the PCGC trios including identical parameters and thresholds.

DNVs (Li et al, 2012; Ramu et al, 2013; Wei et al, 2015), other methods have relied on machine learning techniques (Liu et al, 2014; Khazeeva et al, 2022). However, both approaches have their limitations, which generally require tuning of parameters and/or developing a model based on training sets. Although most of these methods achieve a relatively high sensitivity, the noise present in the WGS data often leads to a significant number of false-positive calls and, as a result, low precision (Liang et al, 2019; Khazeeva et al, 2022; Brand et al, 2024).

Khazeeva et al performed a rigorous work on comparison of performance of DNV callers including GATK, DeepTrio, DeNovoGear, and DeNovoCNN for the HG002-3-4 trio (Table 5) (Khazeeva et al, 2022). DeNovoCNN showed the best performance among the four methods with a sensitivity of 90.6% and a precision of 97.2%, where the number of false-positive calls equaled 35. DeepTrio achieved the second-highest precision at 93.4%, with a sensitivity of 85.2% and 80 false-positive variants. GATK4 called 167 false-positive DNVs and, thus, demonstrated the lowest precision of 87.5%. For comparison, the DeepTrio method in the current study achieved much better performance with a sensitivity of 94.3% and a precision of

97.5%, which increased to 99.0% after the force-calling step (Table 5). All four GATK4 versions (with/without posteriors, with/without the force-calling step) we performed in this study also demonstrated higher precision compared with the GATK4 results reported by Khazeeva et al (2022). We believe that the differences in variant calling stem mostly from our population, regional, and force-calling filters.

In a recent publication, Brand et al retrained and ran DeepTrio on the WGS trios sequenced by the West German Genome Center to call DNVs (Brand et al, 2024). To evaluate performance, they also ran DeNovoCNN and the modified DeepTrio network on the HG002-3-4 Ashkenazi family using it as a reference. They estimated the sensitivity and precision of DeNovoCNN to be 97.0% and 84.1%, respectively, with 182 false-positive variants called (Table 5). The retrained DeepTrio exhibited a sensitivity of 97.8% and a precision of 93.5%, corresponding to 68 false-positive calls (or a sensitivity of 89.6% and a precision of 95.7% depending on the cutoff threshold). Although both methods demonstrated sensitivity exceeding 94.8% found in the current work, our approach achieved a significantly higher precision of 99.2% for the same trio with only eight false-

positive variants. Another notable contribution reported by Brand et al was the detection of clustered de novo variants occurring within 20 base pairs of each other. They noted that identifying these cluster DNVs is more challenging and decreases their estimated sensitivity of the modified DeepTrio model. We did not observe any difficulties in finding cluster DNVs with our workflow. We found nine clusters of DNVs in the HG002-3-4 trio in total including the five mentioned by Brand et al (2024). The length of the clusters of DNVs varied between 2 and 11 base pairs (Table S7). We also found 10 clusters of DNVs with a length of 2 to 17 base pairs in the PCGC dataset (Table S7).

Recently, the consensus approach between GATK4 and Deep-Variant as part of the NVIDIA Parabricks program has been successfully applied to the 1000 Genomes Project, the Simons Simplex Collection, and Simons Foundation Powering Autism Research trio datasets (Ng et al, 2022; Ng & Turner, 2024). The precision of the two-method approach was estimated as 93.6% based on manual inspection of ~4,000 candidate DNVs of the selected four trios, which is lower than 99.5% we observed for a combination of GATK4 and DeepTrio in the HG002-3-4 trio. The difference in precision is probably attributable to the population, regional, and force-calling filters we applied. Notably, for the manual validation of the ~4,000 candidate DNVs, Ng et al used the SAMtools tview (Danecek et al, 2021) to reveal the alternate alleles in parents' pileups and assigned 4.9% of the variants as inherited. In the current study, we performed the same validation automatically by means of the AAHP filter at the force-calling step, which can discover and remove any variants with a few alternate alleles in parents.

Overall, our work advances the consensus methodology by incorporating pangenome analysis and introducing a sequence of filters, which efficiently remove false-positive variants. Our workflow does not require additional parameterization or tuning and allows automated processing of large numbers of trio GS to call DNVs without needing to undertake visual inspections of the BAM files. Given GS data of a particular average read depth and quality, secondary analyses done by different read aligners and variant callers produce notably different lists of DNVs (Fig 1, central, Fig S7, central). Combining results of three independent methods, we filter out Mendelian violations caused by sequencing or alignment errors. Only DNVs found by at least two methods were considered further, significantly reducing the number of DNV candidates (the *Consensus* step in Fig 1, central). Although we filtered out a small number of TP DNVs, as three DNVs from the Freeze set in the PCGC trios (Fig 2A and B) and 40 DNVs from the truth set in the GIAB family (Fig S8A and B), we removed most of FP variants, giving preference to precision over sensitivity. The percentage of the Freeze/truth DNVs found after the consensus step between the three methods is estimated as 99.4% and 95.0% for the PCGC and GIAB trios, respectively (Fig 1, left, Fig S7, left). Notably, relaxing thresholds at the first QC step performed individually for each method can increase sensitivity, but the obvious trade-off is decreased precision.

The final step of our workflow consisted of the recalling of variant candidates in trios and applying the AAHP, HP, and second QC filters (the *force-calling* step in Figs 1 and S7), which were designed to remove FPs persistent within the three methods, mostly alignment errors. Using any combination of the two methods did not affect precision but notably decreased sensitivity (Tables 1 and 5). For the

10 PCGC trios studied, the combination of GRAF and DeepTrio revealed 23 fewer DNVs than the three-method workflow, whereas the combinations of GATK4 and GRAF and GATK4 and DeepTrio missed 52 and 65 DNVs, respectively. Therefore, in the case of limited computational resources, the two-method approach combining the GRAF and DeepTrio pipelines is the most efficient and least expensive (Table 1). At the same time, applying the force-calling filters requires few computational resources and efficiently removes false-positive calls even for individual pipelines (Figs 2 and S8). We expect that the addition of other independent methods (i.e., implementing a workflow with four or more methods) would slowly increase the number of DNVs called and therefore sensitivity, but require increasing resources to run. Compared with the previous PCGC study, which used a different approach including the convolutional neural network trained on manually curated IGV plots (Richter et al, 2020), our current pipeline identified 99.4% of those DNVs and found 5.7% additional DNVs (Table 1). Sanger sequencing validated 23 of the novel DNVs, confirming SNVs and indels at rates of 86.4% and 100%, respectively (Table 2, Fig 1, left). Interestingly, all 23 confirmed novel DNVs were called by the GRAF pipeline, whereas DeepTrio and GATK4 calls contained 22 and 13, respectively (Table S4). Furthermore, GRAF had the lowest number of false-negative variants compared with DeepTrio and GATK4 in the GIAB trio results (Table 5). This agrees with our initial hypothesis that pangenome-aware variant calling methods have the potential to improve discovery of DNVs, as they can call variants in complex genomic regions (Olson et al, 2022).

Notably, we focused on only the de novo calls where parents had homozygous reference genotypes and proband had heterozygous alternative genotypes because our primary goal is to identify pathogenic de novo mutations. Healthy parents are expected to have reference alleles at the same location where there is a de novo variant causing disease in the proband. Therefore, we applied a genotype filter to keep only such de novo candidates and determined the quality thresholds accordingly for all callers in the first QC step. We verified that the presence of reads with alternative alleles in parents is a strong indicator of inherited variants, even when the parental genotype is a homozygous reference. A variant detected in a related sample acts as a robust prior for a putative variant allele, even when the evidence within the sample itself is limited. Of note, there were two exceptions to the rule: one alternate allele in parental reads for SNVs and one alternate allele in parental reads for single nucleotide indels. The former was confirmed by Sanger sequencing in the PCGC trios (Table S5) and comparing with known DNVs for both datasets. The latter cases were found in the truth set of the GIAB trio. As our AAHP filter was designed to remove all indels with alternate alleles in parents, two such truth indels were filtered out during the force-calling step for the HG002-3-4 family (Fig S8B and D). This underscores the significance of integrating pileup information from related samples into secondary analysis for de novo variant filtering and overall variant calling, which we successfully implemented by the force-calling step here.

Applying the three-method workflow, we found 62.5 de novo SNPs and 5.8 de novo indels per proband, consistent with the expected 44–82 de novo SNVs and 3–9 de novo indels per individual (Acuna-Hidalgo et al, 2016; Goldmann et al, 2019). Interestingly, the

0.4 coding de novo SNVs per proband observed in the PCGC trios are fewer than the expected 1–2 per individual. However, this difference is offset by an increased number of DNVs in the UTRs (Table 4C). Coding DNVs are an important cause of Mendelian genetic diseases associated with DNVs as they disrupt or alter gene functions (Gilissen et al, 2014; Iossifov et al, 2014; Deciphering Developmental Disorders Study, 2017). However, they do not explain all the cases. For example, studies of severe, undiagnosed development disorders in children showed that only 42% of individuals carry pathogenic DNVs in coding sequences (Deciphering Developmental Disorders Study, 2017). The contribution of noncoding DNVs to diseases remains to be explored, and we expect that the method presented here will encourage such studies because of its automated and scalable processing and very high precision and sensitivity of the results.

To summarize, the implemented workflow provides a simple and flexible way to investigate DNVs in trios. It retrieves a robust set of DNVs from thousands of variant candidates and efficiently filters out Mendelian violations caused by alignment or sequencing errors without requiring manual inspection of variants, thus enabling scalable analysis of large datasets of trio GS.

# Materials and Methods

We used GS data from 10 parent–offspring trios from the PCGC. The approach for DNA extraction and GS for the trios has been previously described (Richter et al, 2020). In brief, paired-end, short-read genome sequencing was performed with HiSeq X 10 System (Illumina Inc.) and we achieved an average coverage of 35-40x for all samples. To analyze the GS trio data, we ran three analytic pipelines to call de novo SNVs and indels: GATK4 (https://github.com/broadinstitute/gatk) and DeepTrio (https://github.com/huxiaoti/deeptrio), which rely upon alignment to the single haplotype reference genome assembly using BWA-MEM (Li, 2013 Preprint), and GRAF, which uses alignments to a pangenome reference. The tools and methods presented in this study are available on all Velsera cloud platforms. Figs 1 and S1 and Table S1 show all steps performed in the analysis. Human reference genome assembly GRCh38 (Schneider et al, 2017) was used as the basis for variant calls in all pipelines. Although all pipelines require the paired-end reads aligned or not aligned as inputs, gVCF or VCF files of GATK4 and DeepTrio are already available for some datasets as they are often created by default and can be used as input as well. If starting from the paired-end read files, all pipelines generate gVCF and family VCF files for each trio allowing revisiting of variants if needed.

## GATK4 pipeline

The GATK4 pipeline was constructed following the latest version of the Broad Institute's Best Practices Workflow for germline short variant discovery (Fig 1, Table S1), a standard approach to small variant calling with linear reference genomes (DePristo et al, 2011). Paired-end reads were mapped using BWA-MEM followed by variant calling with HaplotypeCaller (Poplin et al, 2018 Preprint). gVCF files generated for each family member were jointly genotyped using GATK GenotypeGVCFs. Variant Quality Score Recalibration and Genotype Refinement steps were applied next. Possible de novo calls were annotated with VariantAnnotator.

## DeepTrio pipeline

DeepTrio is a machine learning–based variant caller that analyzes family trio alignments together (Kolesnikov et al, 2021 Preprint). It employs deep convolutional neural networks to learn variant context and de novo rate from trio data and then uses this model to call variants using trio alignments. We used BWA-MEM–generated alignments of the family trio as input to the DeepTrio variant caller (Fig 1, Table S1), and the resulting gVCF files for all three family members were jointly genotyped using GLnexus (Yun et al, 2021).

## GRAF pipeline

Velsera GRAF Germline Workflow uses a pangenome reference for incorporating genomic variation in the secondary analysis process, enabling reduced reference bias (Supplemental Data 1—Graph Pangenome Reference). In this work, the GRAF pipeline was used with a human pangenome reference incorporating genetic variation posited by large studies of diverse cohorts (Mills et al, 2006; Katsnelson, 2010; Mallick et al, 2016). The paired-end reads from mother, father, and the proband were mapped using the GRAF Aligner to the pangenome reference, and the GRAF VariantCaller was used for calling variants (Figs 1, S2, and S3). The VCF files for trio members were merged.

## HG002-3-4 trio

The same workflow, including parameters and thresholds, was run for the HG002-3-4 Ashkenazi trio, gold-standard GIAB family with the established truth set of DNVs for high-confidence regions (Fig S7) (Baid et al, 2020 Preprint).

# Data Availability

All sequencing data used in this study can be found in the database of Genotypes and Phenotypes (dbGaP; https://www.ncbi.nlm.nih.gov/gap/) under the accession number phs001138. Individual dbGaP IDs are shown in Table S8. All data for the HG002-3-4 Ashkenazi trio are available for public (https://human-pangenomics.s3.amazonaws.com/index.html?prefix=publications/PANGENOME_2022/DeepTrio/samples/). The tools and methods presented in this study are available on all Velsera cloud platforms. The graph-based sequencing data analysis tools used in this study (Velsera GRAF) are freely available to all researchers on all Velsera academic cloud platforms (Cancer Genomics Cloud, Cavatica, BioData Catalyst). To access academic platforms, please reach out via the respective website. The source code of the graph reference construction method is not publicly available. Restrictions apply for commercial use (please contact Velsera for terms and other details). GATK and DeepTrio are also available on GitHub (https://github.com/broadinstitute/gatk, https://github.com/google/deepvariant).

# Supplementary Information

# Acknowledgements

We would like to acknowledge Amanda McPartland for her assistance with the Sanger confirmations. This work was supported by the National Institutes of Health (NIH) [grant numbers: U01HL153009, 5U01HL128711, U01HL098147]. This work was also supported in part through the computational and data resources and staff expertise provided by Scientific Computing and Data at the Icahn School of Medicine at Mount Sinai and supported by the Clinical and Translational Science Awards (CTSA) grant UL1TR004419 from the National Center for Advancing Translational Sciences. Research reported in this publication was also supported by the Office of Research Infrastructure of the National Institutes of Health under award numbers S10OD026880 and S10OD030463. The content is solely the responsibility of the authors and does not necessarily represent the official views of the National Institutes of Health.

## Author Contributions

M Shadrina: data curation, formal analysis, investigation, visualization, methodology, and writing—original draft, review, and editing.
Ö Kalay: conceptualization, investigation, methodology, and writing—original draft.
S Demirkaya-Budak: validation and investigation.
CA LeDuc: investigation and methodology.
WK Chung: conceptualization, resources, supervision, funding acquisition, investigation, methodology, and writing—review and editing.
D Turgut: supervision and visualization.
G Budak: data curation and visualization.
E Arslan: validation and methodology.
V Semenyuk: software.
B Davis-Dusenbery: resources, supervision, and funding acquisition.
CE Seidman: conceptualization, funding acquisition, and writing—review and editing.
HJ Yost: conceptualization, funding acquisition, and writing—review and editing.
A Jain: conceptualization, supervision, project administration, and writing—review and editing.
BD Gelb: conceptualization, supervision, funding acquisition, project administration, and writing—original draft, review, and editing.

## Conflict of Interest Statement

Ö Kalay, S Demirkaya-Budak, D Turgut, G Budak, E Arslan, V Semenyuk, B Davis-Dusenbery, and A Jain were employees of Velsera Inc. throughout the study period. Other authors have no conflict of interests related to this publication.

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
