## [Reviewer comments · Life Science Alliance]

Life Science Alliance

Efficient Identification of de novo Mutations in Family Trios: A Consensus-Based Informatic Approach

Mariya Shadrina, Ozem Kalay, Sinem Demirkaya-Budak, Charles LeDuc, Wendy Chung, Deniz Turgut, Gungor Budak, Elif Arslan, Vladimir Semenyuk, Brandi Davis-Dusenbery, Christine Seidman, H. Yost, Amit Jain, and Bruce Gelb

DOI: <https://doi.org/10.26508/lsa.202403039>

Corresponding author(s): Bruce Gelb, Icahn School of Medicine at Mount Sinai

Review Timeline:	Submission Date:	2024-09-11
	Editorial Decision:	2024-09-12
	Revision Received:	2025-02-14
	Editorial Decision:	2025-02-18
	Revision Received:	2025-03-19
	Accepted:	2025-03-20

Transaction Report:

Please note that the manuscript was previously reviewed at another journal and the reports were taken into account in the decision-making process at *Life Science Alliance*. Since the original reviews are not subject to Life Science Alliance's transparent review process policy, the reports and author response cannot be published.

September 12, 2024

Re: Life Science Alliance manuscript #LSA-2024-03039-T

Dr. Bruce D. Gelb
Mount Sinai School of Medicine
Pediatrics
One Gustave L. Levy Place
Box 1040
New York, NY 10029

Dear Dr. Gelb,

Thank you for submitting your manuscript entitled "Automated Identification of Germline de novo Mutations in Family Trios: A Consensus-Based Informatic Approach" to Life Science Alliance. We invite you to submit a revised manuscript addressing the Reviewer comments.

Thank you for this interesting contribution to Life Science Alliance. We are looking forward to receiving your revised manuscript.

Sincerely,

B. MANUSCRIPT ORGANIZATION AND FORMATTING:

February 18, 2025

RE: Life Science Alliance Manuscript #LSA-2024-03039-TR

Prof. Bruce D. Gelb
Icahn School of Medicine at Mount Sinai
Pediatrics
One Gustave L. Levy Place
Box 1040
New York, NY 10029

Dear Dr. Gelb,

Thank you for submitting your revised manuscript entitled "Efficient Identification of de novo Mutations in Family Trios: A Consensus-Based Informatic Approach". We would be happy to publish your paper in Life Science Alliance pending final revisions necessary to meet our formatting guidelines.

- please be sure that the authorship listing and order is correct
- please upload your main manuscript text as an editable doc file
- please upload your main and supplementary figures as single files
- please add an Abstract and a Summary Blurb/Alternate Abstract to our system
- please add the X and Bluesky handles of your host institute/organization as well as your own or/and one of the authors in our system
- please note that the titles in the system and manuscript file must match
- please remove the page "Content" after the title page in the manuscript file
- please remove figures from the manuscript text and upload them separately
- please incorporate the supplementary references into the main references in the manuscript file
- please upload your Tables in editable .doc or Excel format
- please add your main, supplementary figure, and table legends to the main manuscript text after the references section
- please add an Author Contributions section to your main manuscript text, and in our system
- there is a call-out in the manuscript text for Figure 1C, and this figure doesn't have panel C...please correct
- please add callouts for Figure S2A-B and table S6 to your main manuscript text;

LSA now encourages authors to provide a 30-60 second video where the study is briefly explained. We will use these videos on social media to promote the published paper and the presenting author (for examples, see <https://docs.google.com/document/d/1-UWcFbE4pGcDdcgzcmiuJI2XMBJnxKYeqRvLLrLSo8s/edit?usp=sharing>). Corresponding or first-authors are welcome to submit the video. Please submit only one video per manuscript. The video can be emailed to contact@life-science-alliance.org

A. FINAL FILES:

-- Summary blurb (enter in submission system): A short text summarizing in a single sentence the study (max. 200 characters including spaces). This text is used in conjunction with the titles of papers, hence should be informative and complementary to the title. It should describe the context and significance of the findings for a general readership; it should be written in the

present tense and refer to the work in the third person. Author names should not be mentioned.

B. MANUSCRIPT ORGANIZATION AND FORMATTING:

Sincerely,

March 20, 2025

RE: Life Science Alliance Manuscript #LSA-2024-03039-TRR

Prof. Bruce D. Gelb
Icahn School of Medicine at Mount Sinai
Pediatrics
One Gustave L. Levy Place
Box 1040
New York, NY 10029

Dear Dr. Gelb,

Thank you for submitting your Methods entitled "Efficient Identification of de novo Mutations in Family Trios: A Consensus-Based Informatic Approach". It is a pleasure to let you know that your manuscript is now accepted for publication in Life Science Alliance. Congratulations on this interesting work.

DISTRIBUTION OF MATERIALS:

Again, congratulations on a very nice paper. I hope you found the review process to be constructive and are pleased with how the manuscript was handled editorially. We look forward to future exciting submissions from your lab.

Sincerely,
